# Compression Stockings Improve Cardiac Output and Cerebral Blood Flow during Tilt Testing in Myalgic Encephalomyelitis/Chronic Fatigue Syndrome (ME/CFS) Patients: A Randomized Crossover Trial

**DOI:** 10.3390/medicina58010051

**Published:** 2021-12-29

**Authors:** C. (Linda) M. C. van Campen, Peter C. Rowe, Frans C. Visser

**Affiliations:** 1Stichting CardioZorg, Planetenweg 5, 2132 HN Hoofddorp, The Netherlands; 2Department of Pediatrics, School of Medicine, Johns Hopkins University, Baltimore, MD 21287, USA; prowe@jhmi.edu

**Keywords:** chronic fatigue syndrome (CFS), myalgic encephalomyelitis (ME), compression stockings, tilt-testing, cardiac output (CO), orthostatic intolerance (OI), cerebral blood flow (CBF), extracranial Doppler

## Abstract

*Background and Objectives:* Orthostatic intolerance (OI) is a clinical condition in which symptoms worsen upon assuming and maintaining upright posture and are ameliorated by recumbency. OI has a high prevalence in patients with myalgic encephalomyelitis/chronic fatigue syndrome (ME/CFS). Limited data are available to guide the treatment of OI in ME/CFS patients. We and others have previously described patient-reported subjective improvement in symptoms using compression stockings. We hypothesized that these subjective reports would be accompanied by objective hemodynamic improvements. *Materials and Methods:* We performed a randomized crossover trial in 16 ME/CFS patients. Each underwent two 15-min head-up tilt table tests, one with and one without wearing knee-high compression stockings that provided 20–25 mm Hg compression. The order of the tests was randomized. We measured heart rate and blood pressure as well as cardiac output and cerebral blood flow (CBF) using extracranial Doppler of the internal carotid and vertebral arteries. *Results*: There were no differences in supine measurements between the 2 baseline measurements. There were no differences in heart rate and blood pressure at either end-tilt testing period. Compared to the test with the stockings off, the mean percentage reduction in cardiac output during the test with compression stockings on was lower, 15 (4)% versus 27 (6)% (*p* < 0.0001), as was the mean percentage CBF reduction, 14 (4)% versus 25 (5)% (*p* < 0.0001). *Conclusion:* In ME/CFS patients with orthostatic intolerance symptoms, cardiac output and CBF are significantly reduced during a tilt test. These abnormalities were present without demonstrable heart rate and blood pressure changes and were ameliorated by the use of compression stockings.

## 1. Introduction

The terms chronic fatigue syndrome and myalgic encephalomyelitis (ME/CFS) describe a complex physical illness characterized by debilitating fatigue, post-exertional malaise (PEM), orthostatic intolerance (OI), pain, cognitive problems, sleep dysfunction and an array of other immune, neurological and autonomic symptoms [1,2,3,4].

Orthostatic intolerance (OI) is a heterogeneous clinical condition in which symptoms worsen upon assuming and maintaining upright posture and are ameliorated by recumbency [1,5]. OI has a high prevalence in patients with ME/CFS [1,6]. Limited data are available to guide the treatment of OI in ME/CFS patients, with variable treatment efficacy reported by different groups [7,8,9,10,11,12,13].

We previously described the effectiveness of knee-high stockings in the daily life of ME/CFS patients, measured with validated questionnaires. The positive effects were mainly present in patients with OI [14].

As many orthostatic intolerance symptoms can be attributed to a reduction in cerebral blood flow (CBF) [1,4,6,15,16,17,18], we hypothesized that wearing knee-high stockings that provided 20–25 mm Hg compression would result in an improvement in CBF in ME/CFS patients. For this purpose, we performed CBF and cardiac output (CO) measurements during tilt-testing with and without compression stockings in 16 ME/CFS patients with OI and a normal heart rate (HR) and blood pressure (BP) response, thus without postural orthostatic tachycardia syndrome (POTS) and without orthostatic hypotension (OH).

## 2. Materials and Methods

A total of 18 ME/CFS patients who consented to treatment with compression stockings for OI symptoms were invited to participate. All patients underwent an extensive history and physical examination to confirm that they satisfied the criteria for the diagnosis of CFS or ME and to exclude other conditions that could account for the diagnosis. In each patient, 2 consecutive short duration (15-min) tilt table tests were performed, one with and one without stockings on. The order of the two tests was randomized. The lower leg compression stockings were CEP Run 2.0 brand, with a closed toe, which provided 20–25 mm Hg compression. HR and BP were continuously measured using finger plethysmography [19]. Tilt testing with measurements of CBF and CO was performed as described previously [6,20,21,22]. Measurements were performed after 15 min of supine rest. Thereafter, subjects were tilted head-up to 70 degrees with measurements being repeated while upright after 10 min. After the first tilt, patients were returned to the supine position, and after 15 min the second series of resting and tilt measurements were taken. The supine resting period started after putting the stockings on or taking the stockings off. None of the patients used medications that could influence heart rate and blood pressure or that have been recommended for clinical management of orthostatic intolerance. All patients had a fluid intake of 2 L or more and had a higher salt intake.

### 2.1. Cerebral Blood Flow (CBF) Measurements: Acquisition and Analysis

Extracranial Doppler measurements were performed supine and at end-tilt, using methods described in previous studies [6,20]. Bilateral internal carotid and vertebral artery Doppler flow velocity frames were acquired by one operator (FCV), using a Vivid-I system (GE Healthcare, Hoevelaken, the Netherlands) equipped with a 6–13 MHz linear transducer. High resolution B mode images, color Doppler images and the Doppler velocity spectrum (pulsed wave mode) were recorded in one frame. At least two consecutive series of six frames per artery were recorded. Frames were recorded in the supine position approximately 5 min before the onset of the tilt period, then while upright starting at the 10-min point. Image acquisition for all 4 vessels at each time point (supine, end-tilt and post-tilt) lasted 3 ± 1 min. Blood flow of the internal carotid and vertebral arteries was calculated offline by an investigator (CMCvC) who was unaware of the patient stocking status. Blood flow in each vessel was calculated from the mean blood flow velocities x the vessel surface area and expressed in mL/minute. Flow in the individual arteries was calculated in 3–6 cardiac cycles and data were averaged. Total cerebral blood flow was calculated by adding the flow of the four arteries.

### 2.2. Doppler Measurements for Cardiac Output (CO) Determination: Acquisition and Analysis

Measurements were performed as described previously [22]. Briefly, the time-velocity integral (VTI) of the aorta was measured using a continuous wave Doppler pencil probe connected to a Vivid I machine (GE, Hoevelaken, The Netherlands) with the transducer positioned in the suprasternal notch. A maximal Doppler signal was assumed to be the optimal flow alignment. With optimal gain and scale settings in the supine position, the same settings were used for the other supine period and both standing periods. At least 2 frames of 6 s were obtained. Echo Doppler recordings were stored digitally. VTI frames were obtained in the resting supine position after acquisition of the CBF Doppler frames and while upright at the end of tilt testing. VTI frame acquisition lasted less than 1 min. We measured the diameter of the outflow tract at an earlier point using echocardiography. The aortic VTI was measured by manual tracing of at least 6 cardiac cycles using the GE EchoPac post-processing software. This was done by one operator (CMCvC). Stroke volumes (SV) were calculated from the VTI and the outflow tract area, corrected for the aortic valve area [23,24]. SVs of the separate cycles were averaged. The cardiac output was calculated from the HR and SV.

The study was carried out in accordance with the Declaration of Helsinki. All ME/CFS participants gave informed, written consent. The use of descriptive clinical data of patients was approved by the medical ethics committee of the Slotervaart Hospital, Amsterdam, The Netherlands, P1450.

### 2.3. Statistical Analysis

Data were analyzed using Graphpad Prism version 8.4.2 (Graphpad software, La Jolla, CA, USA) and SPSS version 21 (IBM USA). All continuous data were tested for normal distribution using the D’Agostino–Pearson omnibus normality test and presented as mean (SD) or as median with the inter quartile range (IQR), where appropriate. Within-group results were compared using the paired *t*-test or the Wilcoxon matched-pairs signed-rank test, where appropriate. The differences between the supine CO stockings off and on and the differences between the supine CBF stockings off and on were expressed as the percent error as outlined by Critchley and Critchley [25]. Furthermore, the difference between supine CBF measurements with stockings off and supine CBF of a previous tilt test (interval up to 7 months), as well as end-tilt CBF measurement with stockings off and end-tilt CBF measurements of a previous tilt test, were expressed as the percent error. We considered a *p*-value of <0.01 to be statistically significant.

## 3. Results

Of the 18 participating ME/CFS patients, one patient had insufficient image quality and was excluded. Another patient was excluded because tilt test measurements without stockings could not be performed due to the rapid onset of severe OI symptoms. The randomization procedure resulted in 9 patients who started with stockings off and 7 patients who started with stockings on. The group included 1 male and 15 females with a mean (SD) age of 44 (13) years. The duration of ME/CFS was 13 (9–28) years. Mean body surface area (BSA) was 1.8 (0.2). During a previous standard tilt test (25 min), 14 patients had a normal HR and BP response. One had classic orthostatic hypotension and one vasovagal syncope during previous testing, but not during the current short tilt test. Table 1 shows the hemodynamic characteristics of the patients during the two tilt tests. Supine measurements did not differ significantly between the two tests. The supine CO measurements of the two periods had a percent error of 2 percent, and the supine CBF measurements of the two periods had a percent error of 5 percent. There was a significant difference in end-tilt cardiac output and cerebral blood flow between the two test periods (both *p* < 0.0001) with a smaller reduction compared to supine measurements when participants were wearing compression stockings. CBF measurements were available from a previous tilt test for 12 of the 16 patients. Supine CBF for the previous test was 578 (504–635) mL/min and end-tilt CBF for the previous test (without any treatment) was 433 (393–443) mL/min. Comparisons of supine values and standing values in these 12 patients to the stockings off results was not significantly different (*p* values 0.96 and 0.79 respectively). The percent error of the supine CBF measurements and end-tilt CBF measurements were 7 and 9 percent, respectively.

Figure 1 shows the supine and end-tilt data for CO with stockings off (panel A), for CO with stockings on (panel B), for CBF with stockings off (panel C) and for CBF with stockings on (panel D). Figure 2A shows the percentage reduction in CO (standing versus supine) in the group with stockings off and on. Compared to the test with the stockings off, the mean percentage reduction in cardiac output during the test with compression stockings on was lower, 15 (4)% versus 27 (6)% (*p* < 0.0001), as was the mean percentage CBF reduction, 14 (4)% versus 25 (5)% (*p* < 0.0001), respectively (Figure 2B).

Figure 3 shows the correlation between the percent reduction in CO and the percent reduction in CBF for both stockings off and stockings on. A clear shift to improvement in hemodynamic measurement is shown by the shift towards lower percent reduction in both CO and CBF. The slope of the two correlation lines is not significantly different (*p* = 0.94).

Figure 4 is an example of CO measurements supine (left panel), at end-tilt with stockings off (middle panel) and at end-tilt with stockings on (right panel). Figure 5 illustrates two examples of CBF measurements. The upper row is from the right carotid artery and the lower is from the left vertebral artery. The left panels are the supine measurements, the middle panels are results from stockings off at end-tilt and the right panels are the results from stockings on at end-tilt.

Legend Figure 4 CO: cardiac output (L/min). Images are suprasternal continuous wave Doppler echography showing time velocity integrals (VTI) and surface area is expressed in centimeters.

Legend Figure 5 LVA: left vertebral artery; RCA: right carotid artery. Images are pulsed wave Doppler echocardiography images of extracranial arteries. Traces are max velocity (outer lining) and mean velocity (second lining). The mean velocity is used to calculate the cerebral blood flow for the vessel.

To evaluate the presence of a carry-over effect from the first to the second of the tilt tests, we compared supine CBF and CO before the first tilt to the supine CBF before the second tilt. The mean CBF for both supine periods was 581; the mean difference between periods was 1.56 (6.45). Similarly, the CO was 3.69 L/min for period 1, and 3.67 L/min for period 2; the mean difference between the two periods was 0.02 (0.09). The differences between periods for CBF and CO were not significant, thereby excluding a carry-over effect.

## 4. Discussion

The use of compression bandages is a time-honored therapeutic approach, recommended as long ago as by Hippocrates, who advised the use of compression bandages for venous thrombosis [26]. The majority of recent studies have focused on the use of compression garments in venous and lymphatic disorders [27]. Limited data are available on the use of compression garments in orthostatic intolerance syndromes like OH, [28,29,30,31,32] POTS, [33,34,35,36] and vasovagal syncope (VVS) [37], whether in isolation or in association with spaceflight [38,39], prolonged bedrest [40], long-haul COVID [41,42] or post-surgery [43,44].

We have recently demonstrated that CO is significantly lower during upright tilt in ME/CFS patients compared to healthy controls, even when the ME/CFS patients have a normal HR and BP response to orthostatic stress [21]. We have also demonstrated that during a hemodynamically normal tilt test, CBF was similar during supine posture but significantly lower in ME/CFS patients than in healthy controls during the orthostatic stress [6]. In the present study, we combined these two types of hemodynamic measurements and studied the effect of wearing compression stockings during the tilt test. The most important finding of this randomized trial in ME/CFS patients is that stockings providing 20–25 mm Hg of compression (class II) resulted in improvement of CO and CBF compared to tilt testing without stockings. This study and the previous study, showing an increased perceived exercise capacity using a questionnaire [14], substantiate the clinical use of compression stockings in ME/CFS patients with OI.

Several observations in the present study warrant further emphasis. First, while we studied a limited number of patients, which could have introduced an inclusion bias, their age, gender, disease duration, and CBF reductions were consistent with our data from a large group of patients [6]. Additionally, in a previous study we reported changes in cardiac index (CI) during tilt testing in ME/CFS patients with a normal HR and BP response [21]. For comparison with that study, we not only calculated CO but also the CI in the present study. Comparison of the CI of the previous study and the present study showed no significant differences. Thus, we are confident that the patient data of the present study are representative for the whole ME/CFS patient population.

Second, although an improvement of CO and CBF was observed while wearing the compression stockings, the reductions in CO and CBF during the standing period of the tilt test remained significant compared to the supine position. Nevertheless, when comparing the percent CO reduction with stockings on in the present study with the percent CI reduction in healthy controls of a previous study [21], a difference in CO/CI reduction during the tilt is observed, being 15% with stockings on versus 8% in healthy controls (without stockings) in the previous study. Similarly, the percent CBF reduction with stockings on in the present study was 14%, while in healthy controls of a previous study [6], the percent CBF reduction was 5%. This suggests that, although the CO and CBF improve with stockings on, compared to stockings off, the CO and CBF reductions do not normalize. Further studies with more extensive compression garments (lower and upper leg compression/compression panties) or with a higher degree of compression [35,45,46] are needed to determine whether this approach will further improve CO and CBF changes in ME/CFS patients.

Third, using a questionnaire in our previous study on compression stockings, the positive and negative responses of wearing compression stockings to a variety of physical activities was variable and dependent on the degree of physical activity in question. In contrast, in the present study we found a uniform hemodynamic improvement when patients wore the stockings. Future work will be able to address whether specific symptoms such as exercise intolerance, pain, cognitive symptoms, lightheadedness or co-morbid disease will be more likely to improve with compression stockings.

Fourth, most prior studies have focused on the use of compression garments in patients with OH, POTS or VVS, as outlined above. We have previously demonstrated that an abnormal CBF reduction may also be present in ME/CFS patients with a normal HR and BP response [6]. It therefore is plausible that in non- ME/CFS populations with OI symptoms, CBF abnormalities can be present despite HR or BP responses to upright posture. Further study can determine whether these patients would also benefit from compression therapy.

Fifth, several methodological considerations need to be addressed: reproducibility, tilt duration, the use of extracranial Doppler, and calculation of differences between supine and standing. Reproducibility was good for both CO and CBF measurements, as percent errors were below 10%. This strengthens the use of these Doppler measurements in clinical practice. In this study, tilt duration was 15 min. In a previous tilt study of CBF, measurements were performed at 12 min and at 22 min [6]. CBF reduction in the patients with a normal HR and BP response at 12 min was 19% and at 22 min 24%. Although the reduction at 22 min was significantly larger than at 12 min, the major change in CBF occurred in the first half of the study. A previous transcranial Doppler study (TCD) of Novak [47] in patients with orthostatic cerebral hypoperfusion syndrome showed that CBF velocity measurements were already abnormal compared to healthy controls in the first minute of the tilt. Similarly, a significant CO reduction is present in healthy controls within 30 s of head-up tilt [48]. Thus, the major hemodynamic changes occur very early after onset of tilting. Given the observation that CBF further decreases during a longer tilt test, it remains to be determined whether compression stockings are as effective during a prolonged period of standing compared to a short period of standing. We expressed the change in CO during the tilt as a percentage reduction instead of an absolute reduction. For the CO calculation of the CO, the aortic valve area is multiplied with the mean VTI. A large number of studies have shown that the left ventricular outflow tract used for this calculation may have an ellipsoid form, leading to underestimation of the aortic valve area [49]. To limit the influence of this underestimation, the percent CO change was used, being independent of the fixed valve area.

Sixth, many studies with a variety of patients and healthy controls have related CBF to CO using interventions like vasoactive drugs, lower body pressure, exercise, postural changes, and blood volume changes; see for a review Castle-Kirszbaum et al. [50]. Thus far, these studies have not included ME/CFS patients, and have not measured subjects during a tilt test. In our study, a significant positive correlation was found between the percent change in CO during the tilt and the percent change in CBF. This was present with or without stockings, with a shift to improved hemodynamics with stockings on. Although the percent reductions in both CO and CBF were lower while wearing stockings, the slope of the relation was not significantly different between stockings on and off, nor did the ratio percent CBF reduction/percent CO reduction change during the two test periods. This indicates that CBF closely follows the CO changes in the setting of a tilt test in these patients. However, we did not measure end-tidal CO_2_, right atrial pressure and catecholamine/sympathetic tone changes [50], nor were we able to measure regional differences in brain perfusion. Thus, further and more extensive studies are needed to assess the nature of these variables. Nevertheless, the majority of studies investigating the relationship between CO and CBF have used TCD to estimate CBF. The major limitation of TCD is that TCD measures flow velocity. For the calculation of cerebral blood flow, a measure of the area of the vessel is needed. Because the vessel area of the large intracranial arteries is dependent on PCO_2_ [51], a change in PCO_2_ may result in a vessel area change and therefore also a change in flow velocity, whether or not the arterial flow changes. The second limitation is that the TCD studies usually report results from one vessel and do not take possible heterogeneity of vessel responses into account. The use of extracranial Doppler with measurements of both internal carotid and vertebral arteries overcomes this potential confounding factor.

Seventh, 93% of the participants in this study were female, slightly higher than the 87% enrolled in our larger CBF study of 429 individuals [6], and likely related to the smaller sample size in this study. In our clinical experience, males have a slightly lower incidence of reporting orthostatic intolerance and less often choose treatment with compression stockings. Further studies with a larger number of males will need to evaluate whether there are differences in response between males and females.

Finally, we did not analyze the heterogeneity of responses of the two internal carotid and two vertebral arteries on the use of the compression stockings. For this purpose, a larger group of patients is mandatory, and this research question needs to be studied in future.

## 5. Conclusions

In ME/CFS patients with OI symptoms, CO and CBF are significantly reduced during a tilt test. These abnormalities were present without demonstrable HR and BP responses to upright stress and were improved while wearing compression stockings. This new finding supports the use of the compression stockings in these patients. Compression stockings may also be beneficial in non-ME/CFS populations with orthostatic intolerance.

## Figures and Tables

**Figure 1 medicina-58-00051-f001:**
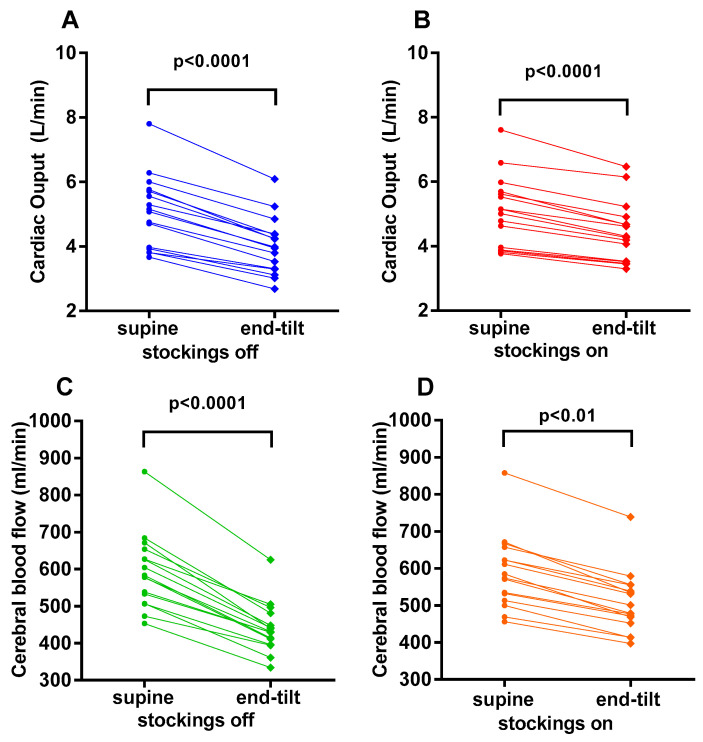
Cardiac output supine and end-tilt for compression stockings off (panel **A**) and compression stockings on (panel **B**). Cerebral blood flow supine and end-tilt for compression stockings off (panel **C**) and compression stockings on (panel **D**).

**Figure 2 medicina-58-00051-f002:**
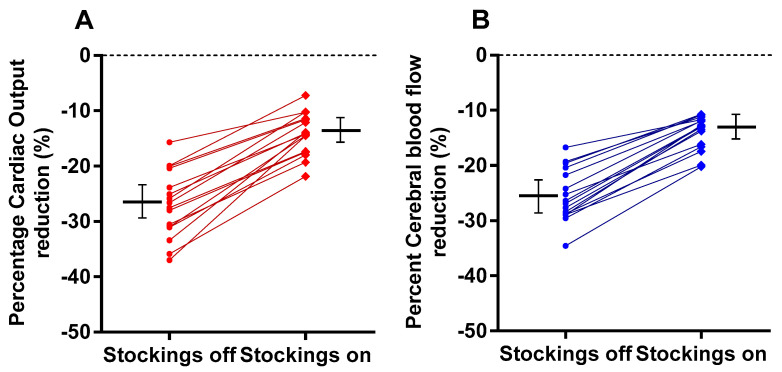
Percentage reduction of cardiac index (panel **A**) and percent reduction of cerebral blood flow (panel **B**) with stockings off (left side of the graph) and stockings on (right side of the graph).

**Figure 3 medicina-58-00051-f003:**
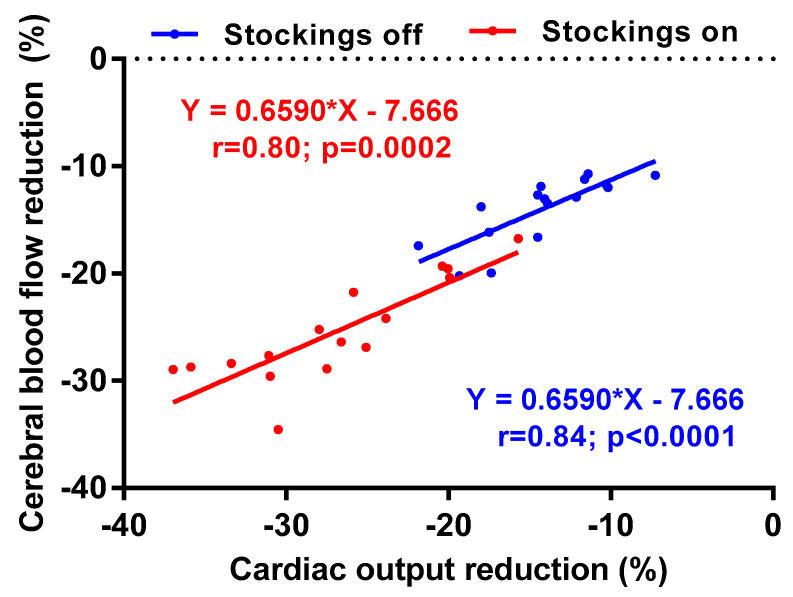
Correlation between the percent reduction in cardiac output and the percent reduction in cerebral blood flow with stockings off (blue) and stockings on (red).

**Figure 4 medicina-58-00051-f004:**
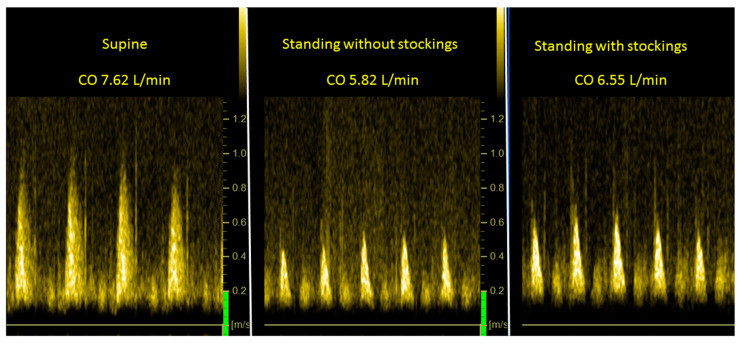
Example of cardiac output supine (**left panel**), at end-tilt without stockings (**middle panel**) and at end-tilt with stockings on (**right panel**).

**Figure 5 medicina-58-00051-f005:**
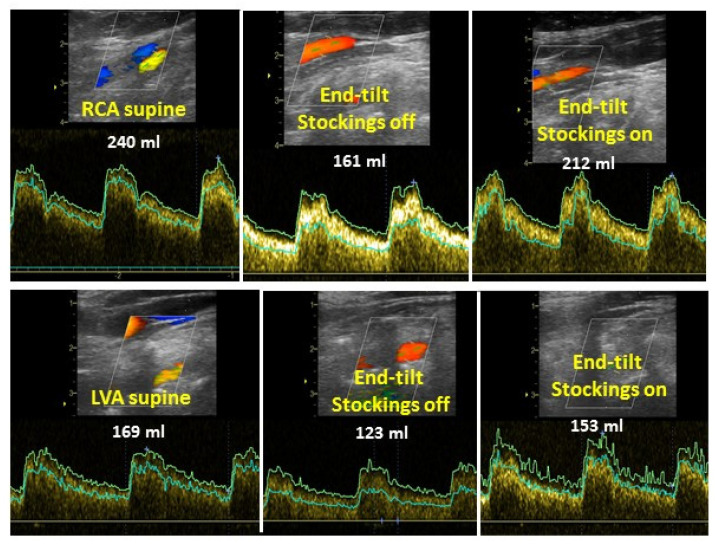
Example of cerebral blood flow supine (**left panel**), at end-tilt without stockings (**middle panel**) and at end-tilt with stockings on (**right panel**). Upper row represents right carotid artery and lower row represents left vertebral artery.

**Table 1 medicina-58-00051-t001:** Hemodynamic baseline characteristics.

	Compression Stockings OFF (*n* = 16)	Compression Stockings ON (*n* = 16)	*p*-Value
Heart rate supine (bpm)	71 (10)	71 (10)	ns
Heart rate upright (bpm)	83 (13)	85 (12)	ns
SBP supine (mmHg)	131 (21)	129 (16)	ns
SBP upright (mmHg)	133 (22)	131 (21)	ns
DBP supine (mmHg)	77 (10)	76 (8)	ns
DBP upright (mmHg)	84 (11)	83 (11)	ns
CO supine (L/min)	5.08 (1.12)	5.06 (1.10)	ns
CO upright (L/min)	4.01 (0.89)	4.43 (0.94)	<0.0001
CBF supine (mL)	580 (513–647)	580 (518–649)	ns
CBF upright (mL)	431 (400–473)	490 (456–551)	<0.0001

SBP = systolic blood pressure, DBP = diastolic blood pressure, CO = cardiac output; CBF = cerebral blood flow.

## Data Availability

The datasets analyzed in the current study are available from the corresponding author on reasonable request.

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
