# Peer review of "Compression Stockings Improve Cardiac Output and Cerebral Blood Flow during Tilt Testing in Myalgic Encephalomyelitis/Chronic Fatigue Syndrome (ME/CFS) Patients: A Randomized Crossover Trial"

_medicina, 2021, doi:10.3390/medicina58010051_

Round 1

Reviewer 1 Report

This is a well-designed trial evaluating an intervention for ME/CFS patients experiencing orthostatic intolerance. Overall, I found the methodology appropriate and the paper clearly written with a good discussion of the implications, strengths, limitations, and future directions. I appreciate the cross-over design which is particularly powerful for smaller sample sizes and the randomization of intervention order/ comparison of supine measurements to account for carry-over effects. The figures and tables were also helpful. I have a few suggestions.

(1) P. 2, lines 46-47. "As orthostatic intolerance symptoms are primarily due to a reduction in cerebral blood flow (CBF) [4, 6, 15-18],......"

I believe it is too premature to make this statement. There are some symptoms like cognitive dysfunction/ "brain fog" that may be due to reduced CBF but others like fatigue, exercise intolerance, nausea, etc. which may be caused mostly or partly by other mechanisms. Indeed, one of the authors' papers suggested there was no relationship between CBF and numerical ratings of fatigue, concentration, and pain. 

https://www.frontiersin.org/articles/10.3389/fmed.2020.602894/full

I do look forward though to reading about whether alleviation of CBF reduction via compression stockings can help address these symptoms. Hopefully, the authors will include not just subjective ratings but objective outcome measures like pain pressure thresholds and N-back cognitive testing as they have done in their other papers. 

(2) Please specify how the study participants were diagnosed with ME/CFS, e.g., which case definitions were used vs. clinical diagnosis by healthcare professional, self-reported diagnosis, etc. The authors can refer to their prior papers if the procedure was similar as before.

(3) Were study participants taking any medications or using any other interventions (e.g., salt/ water loading) that could affect their results? If they were not, additional interventions might improve CBF and CO further. If they were, the results could have been more dramatic. 

(4) P. 7, lines 201-212. Comment further on external validity/ generalizability of results. Even though the demographic characteristics of this sample may be consistent with the authors' clinic population, it is well known that people who are able to access specialty care are often a small group with specific characteristics (e.g., older, wealthier, not people of color, etc.). Also, the inclusion of one man vs. fifteen women is quite different from the known ratios of men : women in ME/CFS. At most 2-4 times more women are affected than men. Finally, the authors should comment on whether study participants were primarily mildly/ moderately affected or if severely affected people were included. 

Author Response

We gratefully thank the reviewer for spending the time on the manuscript. The comments are added.

Reviewer 2 Report

This article by van Campen et al. reported that significant reduction in CO and CBF during a tilt test in ME/CFS patients without HR and BP responses was attenuated while wearing compression stockings, is interesting and would merit publication.

Suggestions are shown as below:

Discussion includes too many unnecessary issues and should be reorganized.

Limitations are too long.

The comparison of the reduction in CO and CBF between the patients in the present study and the controls in the present study is important. The reason why the comparison is methodologically fair is required.

The mechanism or reason why such reductions in CO and CBF were observed in the study patients should be included. Is it possible that they had lower leg muscle pump function? If so, is it primary or secondary to sedentary or non-active life due to the disease?

How do you evaluate the cerebral perfusion? I’m just wondering if enough perfusion was still maintained during upright stress in the study patients.

The important point is that almost identical reduction was observe in CO and CBF during a tilt test in the study patients. If the cerebral perfusion dropped below the demand level enough to maintain orthostasis, autoregulation function should work with lowering cerebral vascular resistance, resulting in milder reduction in CBF than that in CO. In the present study the reduction in CO was the same degree as that in CBF, suggesting no apparent cerebral perfusion disturbance.

Discussion should include the results explanation concerning these issues pointed out.

Author Response

(The authors gave the same response as above.)

Reviewer 3 Report

The manuscript provides therapeutic approaches compression stockings to ameliorated neurovegetative disfunction in ME/CFS patients. It shows a pilot trial but with relevant results for improvement symptomatology in ME/CFS.

  1. The authors should increase the number of ME/CFS patients in the continuing study.
  2. As metioned by authors in methodological considerations, to compare results a standardized method would be better to be used to study O.I. in the continuing study, i.e. , the NASA Lean Test, (NLT) (Lee J, Vernon SD, Jeys P, Ali W, Campos A, Unutmaz D. Hemodynamics during the 10-minute NASA lean test: evidence of circulatory decompensation in a subset of ME/CFS patients. J Trans Med 2020;18:314.).

Author Response

(The authors gave the same response as above.)
